# An Accelerated Failure Time Cure Model with Shifted Gamma Frailty and Its Application to Epidemiological Research

**DOI:** 10.3390/healthcare10081383

**Published:** 2022-07-25

**Authors:** Haro Aida, Kenichi Hayashi, Ayano Takeuchi, Daisuke Sugiyama, Tomonori Okamura

**Affiliations:** 1Graduate School of Science and Technology, Keio University, Yokohama 223-0061, Japan; haro_aida@a5.keio.jp; 2Department of Mathematics, Keio University, Yokohama 223-0061, Japan; 3Department of Predictive Medicine and Public Health, Keio University School of Medicine, Tokyo 160-0016, Japan; ayanotakeuchi@keio.jp (A.T.); okamura@z6.keio.jp (T.O.); 4Faculty of Nursing and Medical Care, Keio University, Fujisawa 252-0816, Japan; dsugiyama@keio.jp

**Keywords:** survival data analysis, accelerated failure time model, cure model, frailty model, epidemiological research

## Abstract

Survival analysis is a set of methods for statistical inference concerning the time until the occurrence of an event. One of the main objectives of survival analysis is to evaluate the effects of different covariates on event time. Although the proportional hazards model is widely used in survival analysis, it assumes that the ratio of the hazard functions is constant over time. This assumption is likely to be violated in practice, leading to erroneous inferences and inappropriate conclusions. The accelerated failure time model is an alternative to the proportional hazards model that does not require such a strong assumption. Moreover, it is sometimes plausible to consider the existence of cured patients or long-term survivors. The survival regression models in such contexts are referred to as cure models. In this study, we consider the accelerated failure time cure model with frailty for uncured patients. Frailty is a latent random variable representing patients’ characteristics that cannot be described by observed covariates. This enables us to flexibly account for individual heterogeneities. Our proposed model assumes a shifted gamma distribution for frailty to represent uncured patients’ heterogeneities. We construct an estimation algorithm for the proposed model, and evaluate its performance via numerical simulations. Furthermore, as an application of the proposed model, we use a real dataset, Specific Health Checkups, concerning the onset of hypertension. Results from a model comparison suggest that the proposed model is superior to existing alternatives.

## 1. Introduction

Survival analysis statistically infers the time until the occurrence of an event of interest, and its main subject of interest is the inference of the survival function. Particularly, investigating how the characteristics of the individuals (covariates) affect survival time has been an epidemiologically important challenge. To tackle this, the proportional hazard model is most commonly used as the regression model for survival time [1]. The advantage of the proportional hazard model is that it allows intuitive interpretation of the results if the covariates are the characteristics obtained at the beginning of the observation of the individuals and are independent of time. In contrast, because the model requires the assumption that the ratio of the hazard functions is independent of time (proportional hazard assumption), its disadvantage is that it may give inappropriate analysis results if the data do not meet the assumption. Furthermore, it has been pointed out that this proportional hazard assumption is strong and often unrealistic [2].

In addition, some of the events of interest may not occur in all individuals (for example, lifestyle-related diseases and the onset of diabetes in AIDS patients). Thus, the proportional hazard model, which assumes that an event always occurs under a sufficiently long observation period, is not suitable for the problem setting mentioned above, and the cure model has been proposed as a model that can be applied to such situations [3,4]. In particular, the mixture cure model divides individuals into latent groups with or without the occurrence of an event (uncured individuals in the case of a curable disease), expressing the survival function as their mixture distribution [5].

To solve these problems, the present study considers the accelerated failure time mixture cure model, which is a flexible model that directly expresses the relationship between the survival function and the covariates and does not necessarily require an assumption regarding the hazard ratios. In this study, we propose the mixture cure model based on the accelerated failure time (AFT) model with frailty, a latent variable that expresses the heterogeneity of the individuals [6], which enables the model to be more flexible. One of our ideas is to assume that the frailty for survival time of the uncured individuals follows a shifted gamma distribution, allowing clearer characterization of uncured individuals. In addition, we analyze the dataset on hypertension using the proposed model and compare its results with those of the existing models. One of the competitive models is a mixture cure model that assumes the proportional hazard model in the survival function for uncured patients. In this case, however, the proportional hazard assumption mentioned above is assumed in the survival function for uncured patients, leading to a model with a strong assumption. On the contrary, the assumption of the AFT model in the survival function for uncured patients allows us to consider a more flexible mixture cure model that does not require an assumption on hazard ratios [7,8].

Our major contribution of the present study is that introducing two aspects of heterogeneity and relaxation of model assumptions in parametric settings. Heterogeneity in this context allows existence of (i) uncured patients (experience an event in the long run) and cured patients in population and (ii) individual difference among uncured patients. Although similar models have been considered in the previous studies (for example, see [9]), our approach is novel in the sense that a shifted gamma frailty is found to improve model fitting and a generalized gamma distribution is considered for the uncured component for flexibility.

The overall organization of this paper is structured as follows: First, Section 2 explains survival time and censoring and defines the functions, such as the survival function, used in survival analysis. It also describes the proportional hazard and AFT models and introduces the survival time regression model considering the cure rate. Section 3 defines the AFT frailty model and the parameter estimation method for frailty. Section 4 describes a numerical example of the proposed model, in which we perform a simulation to evaluate the behavior of estimated values and summarize the results of data analysis on the onset of hypertension for discussion. Finally, Section 5 summarizes this study and discusses future challenges.

## 2. Regression Models for Survival Time Response

### 2.1. Literature Review

We develop a survival model that relaxes some assumptions in the proportional hazard model. Prior to model formulation, we briefly review some related previous studies without showing statistical models. When survival (time-to-even) outcome is of interest, statistical models are often presume that all patients will experience an event if they are observed sufficiently long time. However, this implicit assumption is justified depending on characteristics of the event or context of the study. The cure model considers a fraction of cured patients in a population, and is generally classified into non-mixture and mixture cure models [4]. Since the non-mixture cure model is not suitable for our application, the mixture cure model are considered in our study. While the proportional hazard model is most popular to model a censored survival outcome, it requires a strict “proportionality” assumption on the outcome distribution. Hence, some approach to overcome the restriction the accelerated failure time model becomes popular as an alternative [2,10,11]. To take into account heterogeneity of uncured patients, the concept of frailty is also introduced [6,12]. Frailty is a kind of random effect for each patient; we can interpret it as variability of outcome that does not come from covariates’ effect.

The above reviews are three different generalizations of the proportional hazard model in survival analysis. They have developed independently or integrated partially until recent years [9,13,14,15,16]. In this study, a new model that had all of these elements is proposed with one additional twist: a shifted gamma frailty. We tabulate elements in our proposed model to compare related literature in Table 1.

### 2.2. Problem Formulation

This section provides an overview of the problem setting and the regression model for survival time. Let *T* be a continuous non-negative random variable representing the time of occurrence of an event of interest, and let *C* be a non-negative random variable that represents the censoring time. Let O=min(T,C) and Δ=I(T≤C) be the actual observed survival time (right censoring), and ***Z*** be a *p*-dimensional covariate vector. Based on these, let {O,Δ,Z} be the observed variables. Let the sample size be *n*, with the observed values of the *i*-individual following the probability distribution of ${\left(O,\Delta ,Z^{⊤}\right)}^{⊤}$ independently of each other, and their realization is expressed as ${\left(t_{i},\delta_{i},z_{i}^{⊤}\right)}^{⊤}$. Therefore, the sample is $\left\{\left\{t_{i},\delta_{i},z_{i}\right\},\;i=1,\cdots ,n\right\}$.

Let S(t|z)=P(T>t|Z=z),t>0 be the conditional survival function of *T* given Z=z. In addition,
$$\begin{array}{c} h\left(t|z\right)=\lim_{\epsilon \to 0+}\frac{P(t\leq T<t+\epsilon |T\geq t,Z=z)}{\epsilon } \end{array}$$
is called the conditional hazard function of *T*. Furthermore, because of the expression h(t|z)=−dlogS(t|z)/dt, the relationship between the survival function and the hazard function is as follows: $$\begin{array}{c} S\left(t|Z=z\right)=\exp\left[-\int_{0}^{t}h\left(s|z\right)ds\right] \end{array}$$

The proportional hazard model, proposed by [1], assumes that the conditional hazard function is
$$\begin{array}{c} h\left(t|Z\right)=h_{0}\left(t\right)\exp\left(Z^{⊤}\beta \right),\;t>0 \end{array}$$
where $\beta \in R^{p}$ is the regression coefficient for the covariates, and $h_{0}\left(t\right)$ is the baseline hazard function. The proportional hazard model possesses the property that the ratio of hazard functions between different individuals does not depend on time (proportional hazard assumption). Therefore, for different $Z_{1}$, $Z_{2}$, the following is expressed: (1)$$\begin{array}{c} \frac{h(t|Z_{1})}{h(t|Z_{2})}=\frac{\exp\left(Z_{1}^{⊤}\beta \right)}{\exp\left(Z_{2}^{⊤}\beta \right)}. \end{array}$$

The most well-known regression model for survival analysis is the proportional hazard model, which is often used in actual medical research. The partial likelihood method for the semi-parametric model and the maximum likelihood method assuming a specific distribution for the baseline hazard function can be considered for the estimation of the regression coefficient β [17]. The advantage of the proportional hazard model is that the effects of covariates on the hazard function can be easily interpreted. On the other hand, because this model has a strong assumption that the proportional hazard assumption (Equation 1) always holds, its disadvantage is that it cannot draw appropriate conclusions due to the biased estimation when the true probability distribution violates this assumption.

### 2.3. Mixture Cure Models

It is reasonable to consider that events, such as cancer recurrence or the onset of lifestyle-related diseases, do not occur (or are cured) in some cases. The mixture cure model [3], one of the cure models, takes this into consideration. Let *D* be a random variable that is 1 when it belongs to the event occurrence group and is 0 otherwise. The conditional survival function of *T* can then be expressed as
(2)$$\begin{array}{c} S(t|z)=P(D=1|Z=z)P(T>t|D=1,Z=z)+1\times P(D=0|Z=z). \end{array}$$

The right-hand side of Equation (Equation 2) contains the probability of belonging to the uncured group P(D=1|Z=z), in which the survival function is P(T>t|D=1,Z=z). Let p(Z), be the model for the former and $S_{u}\left(t|Z\right)$ be the model for the latter. The mixture cure model can then be expressed as
(3)$$S\left(t|Z\right)=p\left(Z\right)S_{u}\left(t|Z\right)+\left(1-p(Z)\right),$$
where $\lim_{t\to \infty }S_{u}\left(t|Z\right)=0$, 0≤p(Z)≤1a.e. is assumed. Under the assumption described above, $\lim_{t\to \infty }S\left(t|Z\right)=1-p\left(Z\right)\geq 0$, in which 1−p(Z) can be interpreted as the cured probability.

In general, for the uncured probability p(Z), the logistic regression model with the parameter $\gamma \in R^{p+1}$ is considered to be
(4)$$p\left(Z;\gamma \right)=\frac{\exp\left({\tilde{Z}}^{⊤}\gamma \right)}{1+\exp\left({\tilde{Z}}^{⊤}\gamma \right)},$$
where $\tilde{Z}={\left(1,Z^{⊤}\right)}^{⊤}$. In addition, the proportional hazard and AFT models can be applied to the survival function for uncured patients $S_{u}\left(t|Z\right)$ [5,7]. The observed data likelihood function is maximized in the estimation of model (Equation 3). Let $f_{u}\left(t|Z\right)$ be the probability density function corresponding to $S_{u}\left(t|Z\right)$. For the observed data $\left\{\left(t_{i},\delta_{i},z_{i}\right)\;;\;i=1,\cdots ,n\right\}$, the observation likelihood function *L* of model (Equation 3) is expressed as
(5)$$\begin{array}{cc} L(\beta ,\gamma ,\psi ) & =\prod_{i=1}^{n}{\left[-\frac{\partial }{\partial t}S\left(t_{i}|,z_{i}\right)\right]}^{\delta_{i}}{\left[S\left(t_{i}|,z_{i}\right)\right]}^{1-\delta_{i}}\\ \; & =\prod_{i=1}^{n}{\left[p\left(x_{i}\right)f_{u}\left(t|z_{i}\right)\right]}^{\delta_{i}}{\left[p\left(x_{i}\right)S_{u}\left(t|z_{i}\right)+\left(1-p\left(x_{i}\right)\right)\right]}^{1-\delta_{i}}. \end{array}$$

In addition, the maximization of Equation (Equation 5) can be performed with the EM algorithm (expectation maximization algorithm) [18]. The non-mixture cure model, an approach to the cure model [4], originated from the biological mathematical model of cancer cells. Therefore, its application to the interpretation of data in epidemiological studies, which is the main objective of the present study, is difficult. In fact, the mixture cure model is often applied to clinical trials and observational studies [7,19].

### 2.4. Accelerated Failure Time Models

The AFT model is a regression model that assumes
(6)$$\log\left(T\right)=Z^{⊤}\beta +\xi$$
for *T* [20], with the assumption that ξ is a random variable that represents the error and is independent of ***Z***. Here, let $S_{\varepsilon }\left(t|Z\right)$ be the conditional survival function where $\varepsilon =e^{\xi }$; then,
(7)$$\begin{array}{c} S\left(t|Z\right)=S_{\varepsilon }\left(te^{-Z^{⊤}\beta }\right). \end{array}$$

In addition, using the conditional probability density function $f_{\varepsilon }\left(t\right)$ of ε, the conditional probability density function f(t|Z) of *T* is expressed as
$$\begin{array}{cc} f(t|Z) & =f_{\varepsilon }\left(te^{-Z^{⊤}\beta }\right)e^{-Z^{⊤}\beta }. \end{array}$$

The advantage of the AFT model is that it models the relationship between *T* and ***Z***, and it is equivalent to a linear regression model without censoring. In addition, a model without proportional hazard assumption can also be expressed depending on the distribution that assumes for ξ or ε (it becomes the parametric proportional hazard model in the case of exponential and Weibull distributions). As with the proportional hazard model, there are semiparametric and parametric methods for inferring the AFT model (e.g., [21]). In the present study, we consider the parametric method and make statistical inferences assuming a specific distribution, such as Weibull and logarithmic normal distributions, for ε.

In this chapter, we describe the survival time regression model that incorporates frailty and proposes a model using frailty. Frailty in survival analysis represents heterogeneity that cannot be expressed by covariates. Since Ref. [6] first proposed the frailty model, many have studied the regression models incorporating frailty and their parameter estimation methods (e.g., [13,22]). Furthermore, previous studies, including [14,23], reported cases in which the cure rate was considered. In this chapter, we propose the mixture cure model using the accelerated failure time frailty model that assumes a shifted gamma distribution of frailty in the survival function for uncured patients, and we describe its estimation method.

### 2.5. Frailty Models

Ref. [6] proposed a model incorporating a latent variable, known as frailty, for situations that assume heterogeneity between individuals that cannot be expressed by covariates. Let *Y* be a non-negative random variable that represents frailty and assume that it is independent of (T,Z). The frailty model for the proportional hazard model is thus the conditional hazard function h(t|Z) with
(8)$$h\left(t|Z\right)=Yh_{0}\left(t\right)\exp\left(Z^{⊤}\beta \right).$$

Model (Equation 8) is a normal proportional hazard model when Y=1. In this frailty model (Equation 8), the value of the hazard function increases when Y>1, while it decreases when Y<1. Therefore, it can express the changes in the magnitude of the event occurrence rate for the same ***Z*** depending on the magnitude of *Y*.

Additionally, taking frailty into consideration based on Equations (Equation 6) and (Equation 7), the following can be considered for the AFT model: (9)$$\left\{\begin{array}{c} T=\varepsilon \exp(Z^{⊤}\beta )\\ S_{\varepsilon }\left(t\right)={\left\{\tilde{S}\left(t\right)\right\}}^{Y}, \end{array}\right.$$
where $\tilde{S}\left(t\right)$ is the survival function. Model (Equation 9) is obtained by multiplying the hazard function by a variable representing the frailty from the relationship between the survival function and the hazard function. Based on these, when the covariates and frailty values are given, the survival function is expressed as
(10)$$S\left(t|Z,Y=y\right)=\tilde{S}{\left(te^{-Z^{⊤}\beta }\right)}^{y}.$$

There are two methods of estimation for the AFT frailty model (Equation 8): a method assuming that $\tilde{S}\left(t\right)$ follows a parametric probability distribution [24] and a semiparametric method that does not assume a specific distribution [15].

Since the variable *Y* is a latent variable, we marginalize the conditional survival function S(t|Z,Y) with respect to *Y*. When we denote the distribution function of *Y* by $F_{Y}$, the conditional survival function S(t|Z) of *T* in the frailty model (Equation 8) can be calculated by
(11)$$\begin{array}{cc} S(t|Z) & =\int_{0}^{\infty }S\left(t|Z,Y=y\right)dF_{Y}\left(y\right)\\ \; & =\int_{0}^{\infty }\exp\left[-yH_{0}\left(t\right)\exp\left(Z^{⊤}\beta \right)\right]dF_{Y}\left(y\right), \end{array}$$
where $H_{0}\left(t\right)=\int_{0}^{t}h_{0}\left(s\right)ds$. Similarly, the accelerated failure time frailty model (Equation 9) can be calculated as
(12)$$\begin{array}{cc} S(t|Z) & =\int_{0}^{\infty }S\left(t|Z,Y=y\right)dF_{Y}\left(y\right)\\ \; & =\int_{0}^{\infty }\tilde{S}{\left(te^{-Z^{⊤}\beta }\right)}^{y}dF_{Y}\left(y\right)\\ \; & =\int_{0}^{\infty }\exp\left[-y\tilde{H}\left(te^{-Z^{⊤}\beta }\right)\right]dF_{Y}\left(y\right), \end{array}$$
where $\tilde{H}\left(t\right)$ is a cumulative hazard function corresponding to $\tilde{S}$, and $\tilde{S}\left(t\right)=\exp\left[-\tilde{H}\left(t\right)\right]$.

An example of frailty *Y* is the gamma distribution Gamma(ζ,τ) with the scale parameter ζ and the shape parameter τ. In this frailty model, we assume a one-parameter gamma distribution with ζ=θ, τ=1/θ for identifiability. The probability density function of *Y* is then expressed as
(13)$$f\left(y;\theta \right)=\frac{1}{\theta^{1/\theta }\Gamma \left(1/\theta \right)}y^{1/\theta -1}e^{-y/\theta },\;y>0,\theta >0$$
and E[Y]=1 regardless of the value of θ [25].

Assuming this gamma distribution is the distribution of *Y*, model (Equation 11) is expressed as
(14)$$\begin{array}{c} S\left(t|Z\right)={\left(1+\theta H_{0}\left(t\right)\exp\left(Z^{⊤}\beta \right)\right)}^{-\frac{1}{\theta }} \end{array}$$
and model (Equation 12) obtains
(15)$$\begin{array}{c} S\left(t|Z\right)={\left(1+\theta \tilde{H}\left(te^{-Z^{⊤}\beta }\right)\right)}^{-\frac{1}{\theta }}. \end{array}$$

They are referred to as the gamma frailty model and the AFT gamma frailty model, respectively.

## 3. A Novel Accelerated Failure Time Frailty Mixture Cure Model

Our study uses a shifted gamma distribution for the random variable *Y*. In this section, we address our proposed model, the reason why the above distribution is assumed, and an estimation algorithm.

### 3.1. Proposed Model

The mixture cure model, which is proposed in this section, is based on the survival function $S\left(t|Z\right)=p\left(Z\right)S_{u}\left(t|Z\right)+\left(1-p(Z)\right)$ in Equation (Equation 3). More specifically, we assume the AFT frailty model (Equation 12) in the survival function for the uncured group $S_{u}\left(t|Z\right)$ and the logistic regression model (Equation 4) in the uncured probability p(Z). In addition, this study considers the case in which $\tilde{S}\left(t\right)$ is the survival function of a generalized gamma distribution, and we use a reparametrized representation of the generalized gamma distribution given in the original study by [26]. Therefore, using three parameters μ∈R,σ>0,q∈R, the probability density function is expressed as
(16)$$f_{\mathrm{gg}}\left(x;\mu ,\sigma ,q\right)=\left\{\begin{array}{cc} \left|q\right|\frac{{\left\{q^{-2}\right\}}^{q^{-2}}}{\sigma x\Gamma (q^{-2})}\exp\left[q^{-2}\left(q\frac{\log x-\mu }{\sigma }-\exp\left(q\frac{\log x-\mu }{\sigma }\right)\right)\right] & \left(q\neq 0\right)\\ \frac{1}{\sqrt{2\pi }\sigma x}\exp\left\{-\frac{{\left(\log x-\mu \right)}^{2}}{2\sigma^{2}}\right\} & \left(q=0\right) \end{array}\right.$$
for x>0 [27]. The function (Equation 16) becomes the probability density function of a Weibull distribution when q=1, a gamma distribution when q=σ, and a logarithmic normal distribution when q=0. The generalized gamma distribution is the flexible random model that contains the probability distribution described above, which is often used in survival analysis. Hereafter, the survival function and cumulative hazard function of a generalized gamma distribution are expressed as $S_{\mathrm{gg}}\left(t;\mu ,\sigma ,q\right),\;H_{\mathrm{gg}}\left(t;\mu ,\sigma ,q\right)$, respectively.

Furthermore, we assume a shifted gamma distribution for the frailty *Y* [28]. If $Y^{\prime }∼\mathrm{Gamma}\left(\zeta ,\tau \right)$, then a random variable *Y* that follows a shifted gamma distribution with three parameters, η, ζ, and τ, is defined as
$$Y=\eta +Y^{\prime }$$

Then, E[Y]=η+ζτ and $Var\left[Y\right]=\zeta^{2}\tau$. In this study, the parameter η of a shifted gamma distribution is fixed at η=1, and
(17)$$Y=1+Y^{\prime },Y^{\prime }∼\mathrm{Gamma}\left(\theta ,1/\theta \right)$$
is assumed for the frailty. When the distribution function of *Y* is expressed as $F_{\mathrm{sg}}\left(y\right)$, the marginalized survival function $\overset{ˇ}{S}\left(t|Z\right)$ is as follows: (18)$$\begin{array}{cc} \overset{ˇ}{S}\left(t|Z\right) & =\int_{1}^{\infty }{\left\{\tilde{S}\left(te^{-Z^{⊤}\beta }\right)\right\}}^{y}dF_{\mathrm{sg}}\left(y\right)=\tilde{S}\left(te^{-Z^{⊤}\beta }\right){\left(1+\theta \tilde{H}\left(te^{-Z^{⊤}\beta }\right)\right)}^{-\frac{1}{\theta }}. \end{array}$$

The uncured individuals can be characterized more clearly by assuming this shifted gamma distribution. The reason for this is that, assuming a gamma distribution Gamma(θ,1/θ) for *Y*, it is highly probable that the frailty value becomes 1 or less [29]. That is,
$$\int_{0}^{1}\frac{1}{\theta^{1/\theta }\Gamma \left(1/\theta \right)}y^{1/\theta -1}e^{-y/\theta }dy\geq \frac{1}{2}$$
holds for almost all θ (Figure 1). Therefore, the normal gamma frailty model considers the case in which events are less likely to occur, compared to the case in which frailty is not assumed. In fact, in the analysis of the real dataset discussed below, the estimated value of θ in this model was significantly close to 0, suggesting that the normal gamma frailty model [16] is inappropriate. When the shifted gamma distribution (Equation 17) is assumed for frailty,
$$\overset{ˇ}{S}\left(t|Z\right)=\tilde{S}\left(te^{-Z^{⊤}\beta }\right){\left(1+\theta \tilde{H}\left(te^{-Z^{⊤}\beta }\right)\right)}^{-\frac{1}{\theta }}\leq \tilde{S}\left(te^{-Z^{⊤}\beta }\right)$$
holds for any t∈[0,∞). Therefore, the proposed model can characterize the survival function of the uncured group more clearly compared to the case in which frailty is not assumed. Based on these, the proposed model of the survival rate function is expressed more specifically as
(19)$$\begin{array}{c} S\left(t|Z\right)=p\left(Z;\gamma \right)S_{\mathrm{gg}}\left(te^{-Z^{⊤}\beta };\mu ,\sigma ,q\right){\left(1+\theta H_{\mathrm{gg}}\left(te^{-Z^{⊤}\beta };\mu ,\sigma ,q\right)\right)}^{-\frac{1}{\theta }}\\ \;+\left(1-p(Z;\gamma )\right). \end{array}$$

### 3.2. Estimation Method and Its Algorithm

In this section, we construct the parameter estimation method for the proposed model (Equation 19) based on the EM algorithm. Recall that $\beta \in R^{p},\;\gamma \in R^{p+1}$, and let $\kappa ={\left(\mu ,\sigma ,q\right)}^{⊤}$ and θ be the parameters of the generalized gamma distribution and shifted gamma distribution, respectively.

Let $D_{i}\;\left(i=1,\ldots ,n\right)$ be a random variable that is 1 when the *i*-th individual belongs to the event occurrence group and is 0, otherwise (i=1,…,n). Let the observation time of the *i*-th individual be $O_{i}=\min\left(T_{i},C_{i}\right)$ and the latent sample be $\left\{\left(O_{i},\Delta_{i},Z_{i},D_{i}\right)\;;\;i=1,\cdots ,n\right\}$. The likelihood function L(β,κ,θ,γ) of the proposed model (Equation 19) is expressed as
$$L\left(\beta ,\kappa ,\theta ,\gamma \right)=\prod_{i=1}^{n}{\left\{p\left(Z_{i}\right)f_{u}\left(O_{i}|Z_{i}\right)\right\}}^{\Delta_{i}D_{i}}{\left\{p\left(Z_{i}\right)S_{u}\left(O_{i}|Z_{i}\right)\right\}}^{\left(1-\Delta_{i}\right)D_{i}}{\left\{1-p(Z_{i})\right\}}^{1D_{i}}.$$

Note that indication of parameters on the right-hand side of the equation above is suppressed for simplicity of expression. Here, because
$$f_{u}\left(t|z_{i}\right)=e^{-Z_{i}^{⊤}\beta }f_{\mathrm{gg}}\left(te^{-Z_{i}^{⊤}\beta };\kappa \right){\left(1+\theta H_{\mathrm{gg}}\left(te^{-Z_{i}^{⊤}\beta }\right)\right)}^{-\frac{1}{\theta }-1}\left(2+\theta H_{\mathrm{gg}}\left(te^{-Z_{i}^{⊤}\beta }\right)\right),$$
the log-likelihood function can be written as follows: (20)$$\begin{array}{cc} \ell (\beta ,\kappa ,\theta ,\gamma )\; & =\sum_{i=1}^{n}\left\{\Delta_{i}D_{i}\left[\log f_{\mathrm{gg}}\left(O_{i}e^{-Z_{i}^{⊤}\beta };\kappa \right)-\left(\frac{1}{\theta }+1\right)\log\left(1+\theta H_{\mathrm{gg}}\left(O_{i}e^{-Z_{i}^{⊤}\beta };\kappa \right)\right)\right]\right\}\\ \; & \;+\sum_{i=1}^{n}\left\{\Delta_{i}D_{i}\left[\log\left(2+\theta H_{\mathrm{gg}}\left(O_{i}e^{-Z_{i}^{⊤}\beta };\kappa \right)\right)-Z_{i}^{⊤}\beta \right]\right\}\\ \; & \;+\sum_{i=1}^{n}\left\{\left(1-\Delta_{i}\right)D_{i}\left[\log S_{\mathrm{gg}}\left(O_{i}e^{-Z_{i}^{⊤}\beta };\kappa \right)-\frac{1}{\theta }\log\left(1+\theta H_{\mathrm{gg}}\left(O_{i}e^{-Z_{i}^{⊤}\beta };\kappa \right)\right)\right]\right\}\\ \; & \;+\sum_{i=1}^{n}\left\{D_{i}{\tilde{Z}}_{i}^{⊤}\gamma -\log\left\{1+\exp({\tilde{Z}}_{i}^{⊤}\gamma )\right\}\right\}. \end{array}$$
where ${\tilde{z}}_{i}=\left(1,Z_{i}^{⊤}\right)⊤$. Since the log-likelihood function (Equation 20) is divided into the terms related to {β,κ,θ} and the terms related to γ, their maximization can be performed separately. In the algorithm given below, let the updated value of the *k*-th parameter be $\phi^{\left(k\right)}={\left(\beta^{(k)⊤},\kappa^{(k)⊤},\theta^{(k)⊤},\gamma^{\left(k\right)}\right)}^{⊤}$ and the observed samples be $D=\{\left(O_{i},\Delta_{i},Z_{i}\right)$; i=1,⋯,n}.

To simplify the following notation, $E[\cdot |D;\phi^{\left(k\right)}]$ is written as $\bar{E}\left[\cdot \right]$. If the conditional expected value of the function (Equation 20) is expressed as $\bar{\ell }\left(\beta ,\kappa ,\theta ,\gamma \right)$, the following is then expressed: (21)$$\begin{array}{cc} \bar{\ell }\left(\beta ,\kappa ,\theta ,\gamma \right)\; & =E[\tilde{\ell }\left(\beta ,\kappa ,\theta ,\gamma \right)|D;\phi^{\left(k\right)}]\\ \; & =\sum_{i=1}^{n}\left\{\Delta_{i}\bar{E}\left[D_{i}\right]\left[\log f_{\mathrm{gg}}\left(O_{i}e^{-Z_{i}^{⊤}\beta };\kappa \right)-\left(\frac{1}{\theta }+1\right)\log\left(1+\theta H_{\mathrm{gg}}\left(O_{i}e^{-Z_{i}^{⊤}\beta };\kappa \right)\right)\right]\right\}\\ \; & \;+\sum_{i=1}^{n}\left\{\Delta_{i}\bar{E}\left[D_{i}\right]\left[\log\left(2+\theta H_{\mathrm{gg}}\left(O_{i}e^{-Z_{i}^{⊤}\beta };\kappa \right)\right)-Z_{i}^{⊤}\beta \right]\right\}\\ \; & \;+\sum_{i=1}^{n}\left\{\left(1-\Delta_{i}\right)\bar{E}\left[D_{i}\right]\left[\log S_{\mathrm{gg}}\left(O_{i}e^{-Z_{i}^{⊤}\beta };\kappa \right)-\frac{1}{\theta }\log\left(1+\theta H_{\mathrm{gg}}\left(O_{i}e^{-Z_{i}^{⊤}\beta };\kappa \right)\right)\right]\right\}\\ \; & \;+\sum_{i=1}^{n}\left\{\bar{E}\left[D_{i}\right]{\tilde{Z}}_{i}^{⊤}\gamma -\log\left\{1+\exp({\tilde{Z}}_{i}^{⊤}\gamma )\right\}\right\} \end{array}$$

In addition, $\bar{E}\left[D_{i}\right]$ can be calculated as follows: (22)$$\begin{array}{cc} \bar{E}\left[D_{i}\right] & =E[D_{i}|D;\phi^{\left(k\right)}]\\ \; & =1\times P\left(D_{i}=1|O_{i},\Delta_{i},Z_{i};\phi^{\left(k\right)}\right)+0\times P\left(D_{i}=0|O_{i},\Delta_{i},Z_{i};\phi^{\left(k\right)}\right)\\ \; & =\Delta_{i}P\left(D_{i}=1|\Delta_{i}=1,O_{i},\Delta_{i},Z_{i};\phi^{\left(k\right)}\right)\\ \; & \;+\left(1-\Delta_{i}\right)P\left(D_{i}=1|\Delta_{i}=0,O_{i},\Delta_{i},Z_{i};\phi^{\left(k\right)}\right)\\ \; & =\Delta_{i}+\left(1-\Delta_{i}\right)\frac{p_{\left(k\right)}\left(Z_{i}\right)S_{u,(k)}\left(O_{i}|Z_{i}\right)}{1-p_{\left(k\right)}\left(Z_{i}\right)+p_{\left(k\right)}\left(Z_{i}\right)S_{u,\left(k\right)}\left(O_{i}|Z_{i}\right)}, \end{array}$$
where
$$S_{u,\left(k\right)}\left(t|z\right)=S_{\mathrm{gg}}\left(te^{-z^{⊤}\beta^{\left(k\right)}};\kappa^{\left(k\right)}\right){\left(1+\theta^{\left(k\right)}H_{\mathrm{gg}}\left(te^{-z^{⊤}\beta^{\left(k\right)}};\kappa^{\left(k\right)}\right)\right)}^{-\frac{1}{\theta^{\left(k\right)}}}$$
and $p_{\left(k\right)}\left(z\right)$ is a function in which γ in p(z;γ) is replaced with $\gamma^{\left(k\right)}$.

In the M step, the value of the parameter that maximizes Equation (Equation 21) is obtained based on the calculation result of the E step. As mentioned above, since Equation (Equation 21) is divided into the term related to {β,κ,θ} and the term related to γ, the maximization of the two parameter sets can be performed individually. Therefore, the function $\bar{\ell }\left(\beta ,\kappa ,\theta ,\gamma \right)$ is decomposed into
ℓ¯1(β,κ,θ)=∑i=1nΔiE¯[Di]logfgg(Oie−Zi⊤β;κ)−1θ+1log1+θHgg(Oie−Zi⊤β;κ)+∑i=1nΔiE¯[Di]log2+θHgg(Oie−Zi⊤β;κ)−Zi⊤β(23)+∑i=1n(1−Δi)E¯[Di]logSgg(Oie−Zi⊤β;κ)−1θlog1+θHgg(Oie−Zi⊤β;κ)(24)ℓ¯2(γ)=∑i=1nE¯[Di]Z˜i⊤γ−log1+exp(Z˜i⊤γ)
and the updated value of (k+1)-th parameter can be obtained as follows: $$\begin{array}{cc} \left\{\beta^{\left(k+1\right)},\kappa^{\left(k+1\right)},\theta^{\left(k+1\right)}\right\} & =\underset{\beta ,\kappa ,\theta }{\mathrm{argmax}}{\bar{\ell }}_{1}\left(\beta ,\kappa ,\theta \right)\\ \gamma^{\left(k+1\right)} & =\underset{\gamma }{\mathrm{argmax}}{\bar{\ell }}_{2}\left(\gamma \right). \end{array}$$

The above calculation is organized as an algorithm. Let the sample as the realization *D* be $\left\{\left(t_{i},\delta_{i},z_{i}\right)\;;\;i=1,\cdots ,n\right\}$.

Step 1Set the initial values $\beta^{\left(1\right)},\kappa^{\left(1\right)},\theta^{\left(1\right)},\gamma^{\left(1\right)}$.Step 2Calculate the sample version of Equation (Equation 22) for k=1,2,…. That is,
$$\hat{\bar{E}}\left[D_{i}\right]=\delta_{i}+\left(1-\delta_{i}\right)\frac{p{\left(z_{i}\right)}^{\left(k\right)}S_{u,\left(k\right)}\left(t|z\right)}{p_{\left(k\right)}\left(z_{i}\right)S_{u,\left(k\right)}\left(t|z\right)+p{\left(z_{i}\right)}^{\left(k\right)}S_{u,\left(k\right)}\left(t|z\right)}.$$Step 3Find the updated value for each parameter
$$\begin{array}{c} \left(\beta^{\left(k+1\right)},\kappa^{\left(k+1\right)},\theta^{\left(k+1\right)}\right)=\underset{\beta ,\kappa ,\theta }{\mathrm{argmax}}{\bar{\ell }}_{1}\left(\beta ,\kappa ,\theta \right),\;\gamma^{\left(k+1\right)}=\underset{\gamma }{\mathrm{argmax}}{\bar{\ell }}_{2}\left(\gamma \right). \end{array}$$Step 4If the convergence condition is satisfied, terminate the algorithm and set the estimated values to $\left(\beta^{\left(k+1\right)},\kappa^{\left(k+1\right)},\theta^{\left(k+1\right)},\gamma^{\left(k+1\right)}\right)$. Otherwise, increase the value of *k* by 1 and return to step 2.

The estimators estimated by the EM algorithm are the value that maximizes the observation likelihood and are the maximum likelihood estimated values. Therefore, the estimators estimated by this algorithm have consistency and asymptotic normality under appropriate regularity conditions [30]. The log-likelihood function of the observation likelihood in the proposed model (Equation 19) is expressed as with $\phi ={\left(\beta^{⊤},\kappa^{⊤},\theta ,\gamma^{⊤}\right)}^{⊤}$
(25)$$\begin{array}{cc} \tilde{\ell }\left(\beta ,\kappa ,\theta ,\gamma \right)\; & =\sum_{i=1}^{n}\left[\Delta_{i}\log f\left(O_{i}|Z_{i}\right)+\left(1-\Delta_{i}\right)\log S\left(O_{i}|Z_{i}\right)\right]\\ \; & =\sum_{i=1}^{n}\left[\Delta_{i}\left\{\log p\left(Z_{i}\right)+\log f_{u}\left(O_{i}|Z_{i}\right)\right\}+\left(1-\Delta_{i}\right)\log\left\{p\left(Z_{i}\right)S_{u}\left(O_{i}|Z_{i}\right)+\left(1-p\left(Z_{i}\right)\right)\right\}\right] \end{array}$$

Then, the information matrix of $\phi \in R^{2p+4}$ is expressed as
$$I\left(\phi \right)=-E\left[\frac{\partial^{2}}{\partial \phi \partial \phi^{\prime }}\ell \left(\phi \right)\right]$$
and, from the asymptotic normality of the maximum likelihood estimator, the following holds: $$\sqrt{n}\left(\hat{\phi }-\phi_{0}\right)\overset{d}{\to }N_{2p+4}\left(0,I^{-1}\left(\phi_{0}\right)\right)\;\mathrm{as}\;n\to \infty$$
where $\hat{\phi }={\left({\hat{\beta }}^{⊤},{\hat{\kappa }}^{⊤},\hat{\theta },{\hat{\gamma }}^{⊤}\right)}^{⊤}$ is the maximum likelihood estimator, and $\phi_{0}$ is the true value of ϕ.

## 4. Numerical Examples

In this section, the behavior of estimated values of the parameters of the proposed model is confirmed by simulation. In addition, the dataset on the onset of hypertension will be analyzed using the proposed model and compared with that analyzed using existing models for discussion. The statistical software R was used for all analyses.

### 4.1. Simulations

#### 4.1.1. Setting

The objective of this section is to confirm the behavior of estimated values of the model through a Monte Carlo simulation. First, generate the elements of the random variable vector $Z={\left(Z_{1},Z_{2},Z_{3}\right)}^{⊤}$ representing the covariates follow$$Z_{1}∼\mathrm{Bernoulli}\left(0.5\right),\;Z_{2}∼\mathrm{Unif}\left(0,1\right),\;Z_{3}∼\mathrm{Bernoulli}\left(0.4\right)$$
independently of each other, where Bernoulli(*r*) is a Bernoulli distribution with the parameter r∈(0,1) and Unif(a,b) is a uniform distribution on the interval (a,b). Let *n* be the sample size and $Z_{1},\cdots ,Z_{n}$ be independent copies of ***Z***. Next, for ${\tilde{Z}}_{i}={\left(1,Z_{i}^{⊤}\right)}^{⊤}$ and $\gamma ={\left(\gamma_{1},\gamma_{2},\gamma_{3}\right)}^{⊤}$, let
$$p_{i}=\frac{\exp\left({\tilde{Z}}_{i}^{⊤}\gamma \right)}{1+\exp\left({\tilde{Z}}_{i}^{⊤}\gamma \right)}$$
be the uncured probability of the *i*-th individual and generate $D_{i}∼\mathrm{Bernoulli}\left(p_{i}\right)$ independently of each other. Assuming that the frailty $Y_{i}$ of the *i*-th individual follows the shifted gamma distribution (Equation 17) with the parameter θ independently of $Z_{i}$, the event occurrence time $T_{i}$ for $\beta ={\left(\beta_{1},\beta_{2},\beta_{3}\right)}^{⊤}$ is generated based on
$$T_{i}{|}_{D_{i},Y_{i}}∼\left\{\begin{array}{cc} \infty & \left(D_{i}=0\right)\\ S_{\mathrm{gg}}{\left(te^{-Z_{i}^{⊤}\beta };\kappa \right)}^{Y_{i}} & \left(D_{i}=1\right) \end{array}\right..$$

In addition, it is assumed that the censoring time $C_{i}\;\left(i=1,\cdots ,n\right)$ follows the uniform distribution Unif(0,5) independently of any variables. Based on the variables generated above, $O_{i}=\min\left(T_{i},C_{i}\right)$ and $\Delta_{i}=I\left(T_{i}\leq C_{i}\right)$ are defined, and the observed sample $\left\{O_{i},\Delta_{i},Z_{i}\;\left(i=1,\cdots ,n\right)\right\}$ is obtained to estimate the parameters of the proposed model. In this simulation, the following settings are used for the true values of the parameters:(i)β=(−0.5,0.5,−0.8), (μ,σ,q)=(3,1,2),θ=0.5,γ=(0.1,0.5,−1,0.6)(ii)β=(−0.5,0.5,−0.8), (μ,σ,q)=(3,1,2),θ=3,γ=(0.1,0.5,−1,0.6)

The difference between settings (i) and (ii) is the value of parameter θ of the shifted gamma distribution. We will examine the effects and tendencies of the estimation results due to the difference in the parameter on frailty.

#### 4.1.2. Results

The mean parameter estimate values for settings (i) and (ii) are shown in Table 2 and Table 3, respectively.

For almost all of the estimated values in settings (i) and (ii), the mean approaches the true value, and the standard deviation decreases as the sample size increases. In addition, Figure 2 and Figure 3 show the histograms of estimated values of the main subjects of interest in setting (ii) (n=1000). They show that the estimated values of the regression coefficients $\beta_{1}$ and $\gamma_{1}$ are symmetrically distributed around the true value. A similar result was obtained for σ, a component other than β and γ. Although most of the estimated values of θ are close to their true values, some were far from their true values. In addition, the results suggest that the magnitude of θ affects the parameter *q* of the generalized gamma distribution and that the estimation of θ affects the estimation of *q*. The initial values of all estimations were fixed to the same value in this simulation, but the estimated values were converged to those closer to the true value when the initial values were modified. Therefore, the use of multiple initial values for parameter estimation of the proposed model would be a measure for obtaining an appropriate estimate.

### 4.2. Real Data Example

In this section, we analyze the real dataset on the onset of hypertension using survival time regression models and compare the results.

#### 4.2.1. Dataset and Previous Study

The subject of analysis is the dataset on the Specific Health Checkups and Specific Health Guidance, the program started in April 2008 under the National Health Insurance System to maintain lifestyle-related diseases aged between 40 and 74 years. We analyzed the data from Specific Health Checkups conducted in Habikino City, Osaka Prefecture, Japan. They consist of 8325 residents who participated in Specific Health Checkups in 2008 and were followed until the end of March 2013. Of these, 4993 were females and 3332 were males. In this program, lifestyle information for the participans was collected using a self-report questionnaires about the lifestyle and body measurement values, such as waist circumference, age and body weight, and triglyceride level.

We analyzed the dataset of size 3326 (2202 females and 1124 males) obtained by the same exclusion criteria of [31]. They studied the relationship between the onset of hypertension and lifestyle habits, collected via a standard questionnaire, using the proportional hazard model. The present study used the 14 variables (such as age, waist circumference, eating speed, and amount of drinking) used in this previous study as covariates and considered a maximum of 18 linear models as parameters as it also included categorical variables (refer to the paper for details on the variables).

In the present analysis, the onset of hypertension was regarded as an event of interest, and the regression model was applied for each gender. First, the survival functions for males and females were estimated using the Kaplan–Meier estimator (Figure 4). It showed the five-year survival rates of 0.33 and 0.42 for males and females, respectively, indicating that they were not low. In addition, we performed a hypothesis testing of the proportional hazard assumption based on the Schoenfeld residuals for this dataset [32]. The result indicated that the null hypothesis was rejected for the variable “eating snacks after dinner at least 3 times a week” in the datasets of both males and females at a significance level of 5% (*p*-values for the datasets of males and females were 0.0155 and 0.0203, respectively). Therefore, it was suggested that the survival function for the onset of hypertension does not have the proportional hazard assumption. This result indicates that assuming a model without the proportional hazard assumption is appropriate, considering the probability that an event does not occur.

The following five types of models are applied in this section:Proportional hazard model;AFT model;Mixture cure model with the proportional hazard model;Mixture cure model with the AFT model;Proposed model.

The survival distributions assumed for the model are an exponential distribution and a Weibull distribution for the proportional hazard model. A logarithmic normal distribution and a generalized gamma distribution were used for the AFT model. It should be noted that, based on the definitions of these two models, the model used when an exponential or Weibull distribution is assumed for the baseline hazard function of the proportional hazard model is the same as that used when an exponential or Weibull distribution is assumed as the error variable of the AFT model. A logistic regression model is assumed for all uncured probabilities in the mixture cure model. The distribution mentioned above is used for the survival distribution assumed for the survival function for uncured patients.

In some cases during the analysis using the mixture cure model, the absolute value of the estimated value of the regression coefficient γ in the logistic regression model was extremely large. Therefore, we adopted the method of handling the problem of complete separation by [33], which maximizes the objective function
(26)$$\ell^{*}\left(\gamma \right)=\ell_{\mathrm{logis}}\left(\gamma \right)+\frac{1}{2}\log\left|\hat{I}\left(\gamma \right)\right|$$
where a penalty function is added to the log-likelihood function $\ell_{\mathrm{logis}}\left(\gamma \right)$ of the logistic regression model [34]. For the sample covariates $z_{1},\ldots ,z_{n}$,
(27)$$\hat{I}\left(\gamma \right)=\sum_{i=1}^{n}z_{i}z_{i}^{⊤}\frac{e^{z_{i}^{⊤}\gamma }}{{\left(1+e^{z_{i}^{⊤}\gamma }\right)}^{2}}$$
is an estimated value of the information matrix of $\ell_{\mathrm{logis}}\left(\gamma \right)$. This method was applied to ${\bar{\ell }}_{2}$ for the proposed model (Equation 19). In addition, the Akaike information criterion (AIC) was used as a criterion for comparing the goodness of fit of the model [35].

#### 4.2.2. Results

The regression model used in the analysis and AIC values are shown in Table 4 and Table 5.

As shown in Table 4 and Table 5, the lowest AIC is observed in the proposed model when the variable selection is performed for variables in the logistic regression model in both males and females. The estimated values, 95% confidence intervals, and *p*-values for these estimation results are summarized in Table 6, Table 7, Table 8 and Table 9.

As shown in Table 4 and Table 5, the proposed model with the variable selection of the regression coefficient in the uncured probability had the lowest AIC values. However, other models had lower AIC values than the proposed model when variable selection was not performed. This is likely due to the instability resulting from the 18-parameter logistic regression model in the mixture cure model.

For males, seven variables were selected for the uncured probability after variable selection: “age”, “eating speed: normal”, “eating speed: fast”, “eating snacks after dinner at least 3 times a week”, “increase in body weight by 10 kg or more compared to that at the age of 20 years”, “amount of drinking: occasionally”, and “amount of drinking: less than 1–2 go daily”. Of these, “age”, “eating snacks after dinner at least 3 times a week”, “increase in body weight by 10 kg or more compared to that at the age of 20 years”, and “amount of drinking: less than 1–2 go daily” were judged to have a significant regression coefficient at the 5% significance level. Among the variables included in the survival function for uncured patients, “eating snacks after dinner at least 3 times a week” was judged to have a significant regression coefficient at the 5% significance level. For females, three variables were selected for the logistic regression model of the uncured probability after variable selection: “age”, “eating speed: fast”, and “increase in body weight by 10 kg or more compared to that at the age of 20 years”. Of these, only “age” was judged to be significant at the 5% significance level. There were no variables in the survival function for uncured patients judged to be significant at the 5% significance level. This result suggests that “age” is the only variable that affects the onset of hypertension in females.

Figure 5 shows the probability density function of the estimated shifted gamma distribution. Estimation was also performed when the accelerated failure time gamma frailty model was assumed for the mixture cure model, showing that the estimated values of θ were $9.42\times 10^{-6}$ and $6.93\times 10^{-5}$ for males and females, respectively, which were significantly close to 0. The AIC values were 7006.649 and 11,603.85 for males and females, respectively. These were higher than those observed without the assumption of frailty, suggesting that this model was not appropriate.

The number of variables selected for the logistic regression model differed between males and females, and some variables were judged to be significant only in males, suggesting that the covariates that affect the onset of hypertension differ between males and females. In the distribution of frailty, females had larger estimated values of θ and variance than males. This indicates that females have greater differences in the onset of hypertension among individuals.

## 5. Discussion

In this study, we proposed a model that assumes the AFT frailty model in the survival function for uncured patients in the mixture cure model. The characteristic of the proposed method is that the variable representing frailty is assumed to follow a shifted gamma distribution, which characterizes the uncured individuals more clearly. In addition, we constructed a parameter estimation method for the proposed model using the EM algorithm and confirmed that the estimation could be performed appropriately using a simulation. Furthermore, we analyzed the dataset on the onset of hypertension in epidemiological research. Compared with existing models, the proposed model showed the lowest AIC values for both males and females.

Ref. [31] reported the results of the analysis using a semiparametric proportional hazard model. Since a parametric model was used in the analysis of the present study, the results of these two studies cannot be compared directly using AIC. Therefore, comparison criteria between semiparametric and parametric models need to be considered in the future. Previously, comparison criteria between semiparametric and parametric models in the proportional hazard model were studied by [36,37], who used approaches based on the focused information criterion (FIC).

The model proposed in this study uses the parametric AFT model, but the method using the semiparametric AFT model can be considered as an alternative. The semiparametric AFT frailty model was previously studied by [38]. Because this model can estimate regression parameters without assuming a specific distribution in the error term, a more flexible model can be considered. On the other hand, its estimation method is more complicated than that of the parametric model. One future challenge is to extend the model proposed in this study to the semiparametric model and construct an estimation algorithm.

In addition, while this study constructed the estimation algorithm assuming only right censoring, there are other types of censoring, one of which is interval censoring. Interval censoring refers to the cases in which only the occurrence of an event between two observation periods is recorded [39]; it is a more generalized type of censoring and includes right censoring. The construction of an estimation algorithm for the datasets that contain interval censoring is also needed in the future. While models considered in the study are linear static models, dynamic prediction for survival data is an important issue [40]. In addition, recent development for big data such as machine learning techniques would be incorporated [41].

## Figures and Tables

**Figure 1 healthcare-10-01383-f001:**
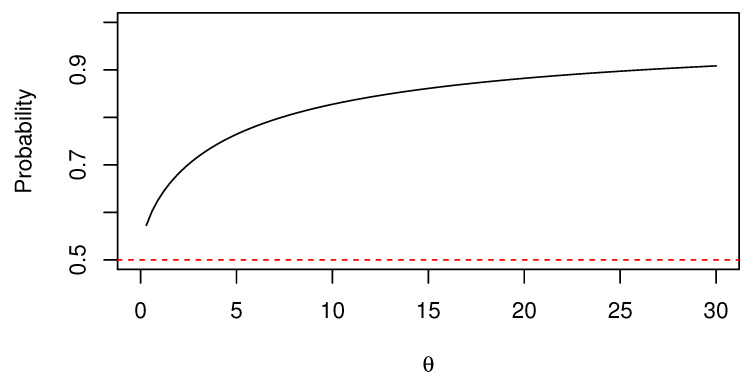
Graph of probability P(Y<1) for Y∼Gamma(θ,1/θ).

**Figure 2 healthcare-10-01383-f002:**
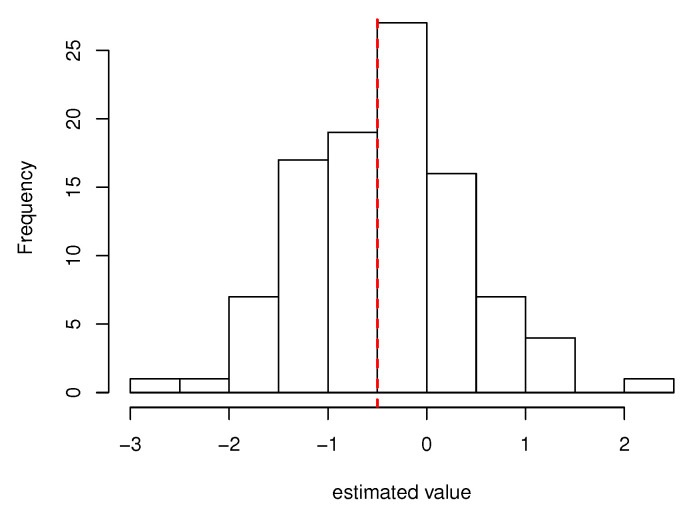
Histogram of the estimated values of $\beta_{1}$ in setting (ii). The red dotted line represents the true value.

**Figure 3 healthcare-10-01383-f003:**
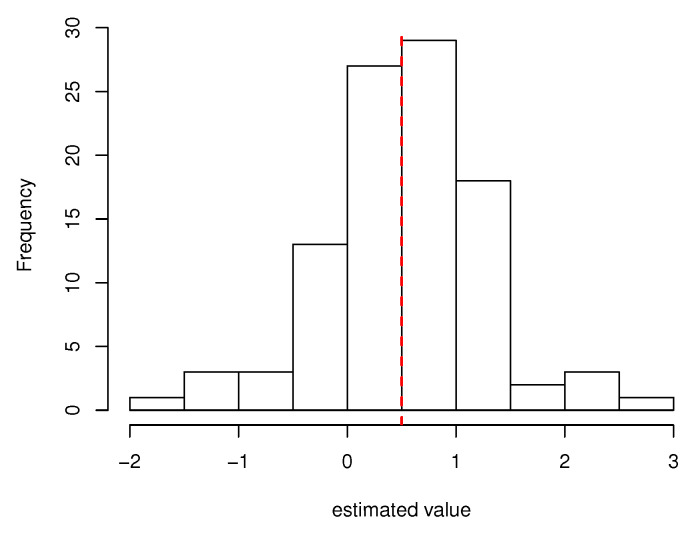
Histogram of the estimated values of $\gamma_{1}$ in setting (ii). The red dotted line represents the true value.

**Figure 4 healthcare-10-01383-f004:**
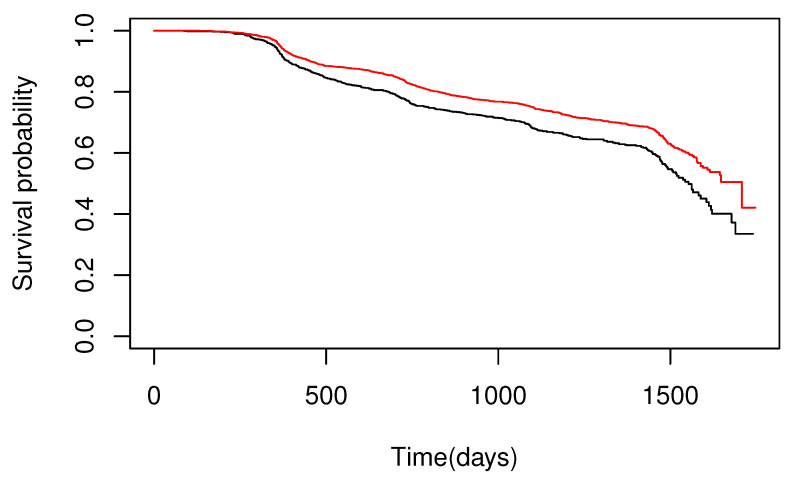
Kaplan–Meier estimator of the survival function for the onset of hypertension for each gender. The black and red lines represent the estimators for males and females, respectively.

**Figure 5 healthcare-10-01383-f005:**
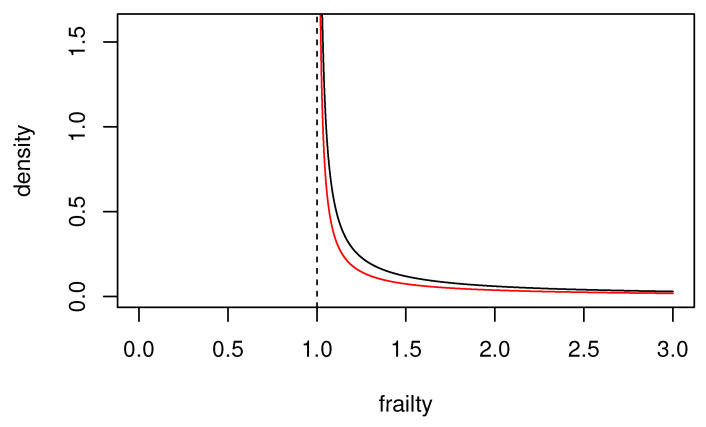
Probability density function of the estimated shifted gamma distribution. The black and red lines show the results for males and females, respectively.

**Table 1 healthcare-10-01383-t001:** Selected related literature and elements of our proposed method. Elements of our proposed method and relationship among existing studies. The symbol ✓ means “Considered”. PH and AFT means proportional hazard and accelerated failure time models, respectively.

Literature	Cured Patients	Uncured Model	Frailty
Sy and Taylor [5]	✓	PH	–
Vaupel [6], Aalen [12]	–	PH	gamma
Pan [13]	–	AFT	gamma, log-normal
Chen et al. [15]	–	AFT	generalized gamma
Yu [9], Price and Manatunga [14]	✓	PH	gamma
He [16]	✓	AFT	generalized gamma
Present study	✓	AFT	shifted gamma

**Table 2 healthcare-10-01383-t002:** Means and standard deviations (SD) of estimate values in setting (i).

Parameter	True Value	Mean (SD)					
		*n* = 100		*n* = 500		*n* = 1000	
$\beta_{1}$	−0.5	−0.355	(1.602)	−0.507	(0.948)	−0.529	(0.803)
$\beta_{2}$	0.5	0.278	(3.000)	0.796	(1.556)	0.814	(1.203)
$\beta_{3}$	−0.8	−0.402	(1.830)	−0.651	(0.925)	−0.656	(0.737)
μ	3	2.091	(4.390)	2.889	(2.074)	2.844	(0.985)
σ	1	1.141	(0.852)	1.004	(0.641)	1.179	(0.743)
*q*	2	1.435	(3.412)	2.804	(1.835)	2.600	(2.012)
θ	0.5	8.214	(20.992)	3.053	(9.309)	2.273	(5.812)
$\gamma_{0}$	0.1	−0.063	(1.526)	0.135	(1.110)	0.192	(0.927)
$\gamma_{1}$	0.5	0.540	(1.216)	0.576	(0.853)	0.553	(0.818)
$\gamma_{2}$	−1	−1.031	(2.294)	−0.878	(1.556)	−0.865	(1.308)
$\gamma_{3}$	0.6	1.045	(1.077)	0.785	(0.852)	0.829	(0.725)

**Table 3 healthcare-10-01383-t003:** Means and standard deviations (SD) of estimate values in setting (ii).

Parameter	True Value	Mean (SD)					
		*n* = 100		*n* = 500		*n* = 1000	
$\beta_{1}$	−0.5	−0.092	(1.796)	−0.316	(1.072)	−0.452	(0.823)
$\beta_{2}$	0.5	0.269	(3.277)	0.552	(1.875)	0.548	(1.285)
$\beta_{3}$	−0.8	−0.386	(1.619)	−0.777	(1.032)	−0.698	(0.904)
μ	3	1.378	(2.993)	3.253	(4.296)	2.840	(1.163)
σ	1	0.910	(0.768)	1.206	(0.878)	1.077	(0.718)
*q*	2	7.276	(49.95)	2.706	(2.197)	2.928	(2.184)
θ	3	23.076	(72.067)	8.874	(21.858)	8.258	(17.101)
$\gamma_{0}$	0.1	−0.093	(1.333)	0.185	(1.335)	0.109	(0.962)
$\gamma_{1}$	0.5	0.699	(1.191)	0.704	(0.926)	0.534	(0.748)
$\gamma_{2}$	−1	−1.159	(2.140)	−1.133	(1.587)	−0.971	(1.141)
$\gamma_{3}$	0.6	0.933	(1.269)	0.608	(0.771)	0.692	(0.844)

**Table 4 healthcare-10-01383-t004:** AIC for the regression model of the onset of hypertension in males. The asterisk (*) represents that variable selection is performed for the uncured probability p(Z).

Model	Distribution	Number of Parameters	AIC
Proportional hazard (PH)	Exponential	19	7120.407
	Weibull	20	7030.615
AFT model	Log-normal	20	7002.352
	Generalized gamma	21	6999.232
Mixture cure + PH	Exponential	38	7103.040
	Weibull	39	7047.378
Mixture cure + AFT	Log-normal	39	7005.633
	Generalized gamma*	40	7004.674
	Generalized gamma *	29	6992.934
Mixtrue cure + AFT frailty	Generalized gamma Shifted gamma*	41	7003.011
	Generalized gamma Shifted gamma *	30	**6987.012**

**Table 5 healthcare-10-01383-t005:** AIC for the regression model of the onset of hypertension in females. The asterisk (*) represents that variable selection is performed for the uncured probability p(Z).

Model	Distribution	Number of Parameters	AIC
Proportional hazard (PH)	Exponential	19	11,804.01
	Weibull	20	11,644.16
AFT	Log-normal	20	11,596.45
	Generalized gamma	21	11,586.00
Mixture cure + PH	Exponential	38	11,798.48
	Weibull	39	11,669.49
Mixture cure + AFT	Log-normal	39	11,609.63
	Generalized gamma *	40	11,600.23
	Generalized gamma *	27	11,579.07
Mixture cure + AFT frailty	Generalized gamma Shifted gamma *	41	11,596.26
	Generalized gamma Shifted gamma *	26	**11,575.05**

**Table 6 healthcare-10-01383-t006:** Estimation result of regression coefficient when variable selection is performed by applying the proposed model in males. CI is the confidence interval. The asterisk (*) and dagger (†) indicate that *p*-value is less than 0.10 and 0.05, respectively.

Covariate	Inference of β (Regression Coefficients for the Uncured Group)		
	Estimates	95% CI	*p*-Value
age	−0.0071	(−0.0188,0.0046)	0.232
waist	−0.0077	(−0.0183,0.0029)	0.155
exe1h_day	−0.0392	(−0.2006,0.1221)	0.634
exe30_2_week	0.0470	(−0.0112,0.0517)	0.562
sleep_good	0.0138	(−0.1732,0.2009)	0.885
walk_speed	−0.1112	(−0.2617,0.0393)	0.148
eat_speed_n	0.1386	(−0.1019,0.3791)	0.259
eat_speed_f	0.0672	(−0.1617,0.2961)	0.565
eat_b_sleep	0.0025	(−0.1920,0.1970)	0.980
snacking	−0.3154	(−0.6195,−0.0112)	0.042 ${}^{†}$
breakfast	−0.1784	(−0.4329,0.0760)	0.169
weight_move	−0.0227	(−0.1956,0.1502)	0.797
plus10kg	0.0162	(−0.1896,0.2219)	0.878
smoking	−0.0673	(−0.2229,0.0884)	0.397
drink_amount2	−0.1636	(−0.4831,0.1558)	0.315
drink_amount3	−0.2134	(−0.4313,0.0045)	0.054 *
drink_amount4	−0.1120	(−0.3288,0.1048)	0.311
drink_amount5	−0.1824	(−0.4299,0.0650)	0.148
**Covariate**	**Inference of γ (Regression Coefficients for the Cured Group)**		
	**Point Estimates**	**95% CI**	***p*-Value**
Intercept	−4.9869	(−7.7813,−2.1924)	0.0005 ${}^{†}$
age	0.0882	(0.0423,0.1340)	0.0002 ${}^{†}$
eat_speed_n	0.8394	(−0.2355,1.9144)	0.1259
eat_speed_f	0.8427	(−0.0571,1.7424)	0.0664 *
kanshyoku	−1.4826	(−2.4026,−0.5626)	0.0016 ${}^{†}$
plus10kg	0.9488	(0.0316,1.8666)	0.0426 ${}^{†}$
drink_amount2	−0.9510	(−1.9441,0.0420)	0.0605 *
drink_amount4	1.1910	(0.0918,2.2901)	0.0337 ${}^{†}$

**Table 7 healthcare-10-01383-t007:** Results of parameter estimation when variable selection is performed by applying the proposed model in males.

Parameter	Distribution			
	Generalized Gamma			Shifted Gamma
	*μ*	*σ*	*q*	*θ*
Estimates	8.4404	0.8767	−0.3141	13.1350
Standard error	0.6150	0.0866	0.2701	7.0537

**Table 8 healthcare-10-01383-t008:** Estimation result of regression coefficient when variable selection is performed by applying the proposed model in females. CI is the confidence interval. The asterisk (*) and dagger (†) indicate that *p*-value is less than 0.10 and 0.05, respectively.

Covariate	Inference of β (Regression Coefficients for the Uncured Group)		
	Estimates	95% CI	*p*-Value
age	−0.0103	(−0.0222,0.0016)	0.089 *
waist	−0.0034	(−0.0100,0.0033)	0.321
exe1h_day	−0.0493	(−0.1717,0.0732)	0.430
exe30_2_week	−0.0230	(−0.1456,0.0995)	0.712
sleep_good	0.0855	(−0.1126,0.1297)	0.890
walk_speed	0.0401	(−0.0726,0.1528)	0.486
eat_speed_n	−0.0651	(−0.2045,0.0742)	0.360
eat_speed_f	0.0520	(−0.1051,0.2092)	0.516
eat_b_sleep	−0.0342	(−0.2395,0.1711)	0.744
snacking	0.0442	(−0.1075,0.1958)	0.568
breakfast	−0.0781	(−0.3625,0.2063)	0.591
weight_move	−0.0210	(−0.1589,0.1169)	0.765
plus10kg	0.0301	(−0.1391,0.1993)	0.727
smoking	−0.1611	(−0.0638,0.3859)	0.160
drink_amount2	−0.0098	(−0.1464,0.1268)	0.888
drink_amount3	−0.0500	(−0.2688,0.1688)	0.654
drink_amount4	−0.1566	(−0.5165,0.2032)	0.394
drink_amount5	−0.2162	(−0.8744,0.4419)	0.520
**Covariate**	**Inference of γ (Regression Coefficients for the Cured Group)**		
	**Point Estimates**	**95% CI**	***p*-Value**
Intercept	−6.0049	(−8.3249,−3.6849)	<0.001 ${}^{†}$
age	0.1045	(0.0645,0.1445)	<0.001 ${}^{†}$
eat_speed_f	0.5600	(−0.1536,1.2736)	0.124
plus10kg	0.6271	(−0.1873,1.4415)	0.131

**Table 9 healthcare-10-01383-t009:** Results of parameter estimation when variable selection is performed by applying the proposed model in females.

Parameter	Distribution			
	Generalized Gamma			Shifted Gamma
	*μ*	*σ*	*q*	*θ*
Estimates	8.1034	0.9653	−0.8065	22.9620
Standard error	0.4688	0.0794	0.2772	9.9425

## Data Availability

Interested readers can contact the corresponding author.

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
