# Peer review of "An Accelerated Failure Time Cure Model with Shifted Gamma Frailty and Its Application to Epidemiological Research"

_healthcare, 2022, doi:10.3390/healthcare10081383_

Round 1

Reviewer 1 Report

# Paper discusses the accelerated failure time cure model with frailty for uncured patients. Also, presents the "frailty" which is a latent random variable representing patients’ characteristics. This is appreciated.

# Add one paragraph on "Major Contributions" of this paper at Second last of Introduction Section.

#Add one paragraph on "overall organization" of this paper at Last of Introduction Section.

# "Literature & Background work" need to be improved in this paper. It is suggested to add one dedicated Section/Sub-section on it after Introduction Section. Also, add one or two comparative tables on different techniques associated to it.

# Cite below articles to improve the readability of this paper:

(a)S. Karthik, RS Bhadoria, JG Lee, AK Sivaraman, S. Samanta, A. Balasundaram, BK Chaurasia, S. Ashokkumar, Prognostic Kalman Filter Based Bayesian Learning Model for Data Accuracy Prediction, Computers, Materials & Continua (CMC), Vol. 72(1):243-259, 2022.

(b) Singh, L. K., Garg, H., Khanna, M., & Bhadoria, R. S. An enhanced deep image model for glaucoma diagnosis using feature-based detection in retinal fundus. Medical & Biological Engineering & Computing, 59(2), 333-353, 2021.

Author Response

We would like to express appreciation to the associate editor and reviews for providing significative feedback on our manuscript. The revised manuscript reflects all suggestions kindly pointed by them. We believe that the presentation of the reviewed manuscript becomes clearer to understand.

Below, we provide each response to all comments.

We added a paragraph addressing major contributions of our study at the second last paragraph of Section 1. We also separate the last paragraph into two parts and set the latter one as the last paragraph of Section 1 to explain the overall organization of the manuscript. In Section 2, as the reviewer recommended, we add two subsections "Literature review" with one table that visualizes the relationship between our study and previous research. In the process of tabulation, we added Yu (2008), Hutton and Monaghan (2002), Aalen (1988), He (2021) as references to reinforce "Literature & Background work". The added table (Table 1) shows the difference of our study and previous ones from the viewpoint of essential elements on survival data analysis. This is also a reflection of a comment from the Reviewer 2.

As the reviewer 1 suggested, we cited two papers (Karthik et al., 2022; Singh et al. 2021) in Section 5. This is because these works focus on prediction of (sometimes not survival) outcome, where nonlinear regression models work well. In our understanding that reviewer's suggestion sheds light on a future direction of our study that we did not recognize. I appreciate the suggestion.

Finally, we uploaded a certification (pdf file) on English editing services below.
https://keio.box.com/s/fy5e4ahy6yxikr4zb7eymebgdd8jvo9z
 We again appreciate the editor and reviewers' useful comments that would improve our manuscript. 

Reviewer 2 Report

I can see the value added of your research and proposed modified survival approach for medical data analysis. In my opinion the missing part is scientific literature review and discussion at the end. You write about weaknesses of traditional survival approach but it is not supported by literature findings. Description of such findings would be value added for your paper.

I strongly suggest to supplement your paper with above mentioned part literature review and discussion description.

Minor comment: on page 3 there is full citation in the text instead of number refernce from refernce list: (Yamaguchi, 1992; Sy and Taylor, 2000).

Author Response

We would like to express appreciation to the associate editor and reviews for providing significative feedback on our manuscript. The revised manuscript reflects all suggestions kindly pointed by them. We believe that the presentation of the reviewed manuscript becomes clearer to understand.

Below, we provide each response to all comments.

We added scientific literature review in the new subsection 2.1 "Literature review" and one possible future direction of the study in Section 5 with two additional references. We understand this was insufficient content of the former version of the manuscript. So we showed Table 1 to clarify the different and similar aspects of our study from previous works. We also discuss further development of the existing approaches of survival analysis required with new references Yu (2008), Hutton and Monaghan (2002), Aalen (1988), He (2021). A mistake on citation in the paper (pointed as a minor comment) is also, of course, fixed.

Finally, we uploaded a certification (pdf file) on English editing services below.
https://keio.box.com/s/fy5e4ahy6yxikr4zb7eymebgdd8jvo9z
 We again appreciate the editor and reviewers' useful comments that would improve our manuscript. 
